# Sensorimotor cortex activity during basketball dribbling and its relation to creativity

**Thomas Kanatschnig**[1,2]\*, **Christian Rominger**[1], **Andreas Fink**[1,3], **Guilherme Wood**[1,2,3], **Silvia Erika Kober**[1,2,3]

**1** Institute of Psychology, University of Graz, Graz, Austria, **2** Complexity of Life in Basic Research and Innovation (COLIBRI), University of Graz, Graz, Austria, **3** BioTechMed-Graz, Graz, Austria

\* thomas.kanatschnig@uni-graz.at

**Data Availability Statement:** All relevant data are within the paper and its Supporting Information files.

**Funding:** The authors acknowledge the financial support by the University of Graz.

## Abstract

Previous studies suggest that it is possible to influence creative performance, by increasing the level of activity in one of the brains hemispheres through unilateral hand movements. Stronger right-hemispheric brain activation due to left-hand movement is assumed to promote creative performance. In this study the aim was to replicate these effects and to expand previous findings, by incorporating a more advanced motor task. 43 right-handed participants were asked to dribble a basketball with the right (n = 22) or left hand (n = 21), respectively. During dribbling the brain activity was monitored over the sensorimotor cortex bilaterally using functional near-infrared spectroscopy (fNIRS). By investigating two groups (left-hand dribbling vs. right-hand dribbling) and by conducting a pre-/posttest design for measuring creative performance (verbal as well as figural divergent thinking tasks), effects of both left- and right-hemispheric activation on creative performance were examined. The results show that creative performance could not be modulated through basketball dribbling. Yet the analysis of the brain activation patterns in the sensorimotor cortex during dribbling revealed findings that largely correspond to the results on hemispherical activation differences during complex motor tasks. Higher cortical activation in the left hemisphere, relative to the right hemisphere, during right-hand dribbling, as well as higher bilateral cortical activation during left-hand dribbling, compared to right-hand dribbling was observed. The results of a linear discriminant analysis further revealed that high group classification accuracy could be achieved using sensorimotor activity data. While we were not able to replicate effects of unilateral hand movements on creative performance, our results reveal new insights into the functioning of sensorimotor brain regions during advanced movement.

## Introduction

Creativity is certainly among the most intriguing subjects of psychological research. The interest in understanding the underpinnings of creative achievements is high and possible real-life implementations of research findings, e.g., in education, the workplace or the arts, have the potential to be beneficial for society [1]. There have been several advancements in creativity research in the past few decades, mainly through the rising use of neuro-technology, such as

**Competing interests:** The authors have declared that no competing interests exist.

electroencephalography (EEG) and functional magnetic resonance imaging (fMRI). Findings from EEG studies draw a convincing connection between cortical processes of creative thinking and oscillations of the EEG alpha frequency band (for an overview see [2]). For example, a prominent finding in this field is that creative performance is associated with higher levels of alpha synchronization, especially at frontal and parietal sites [3–6]. Additionally, studies also reveal hemispheric differences in alpha activity during creative thinking, which corroborates the common notion that both hemispheres support specific cognitive functions during the creative thinking process (e.g., [7]). Research on the brain mechanisms underlying creativity also motivated research to enhance creative performance. This has been done, for instance through interventions such as meditation [8], computerized training [9], stimulation via the exposure to other people's ideas [10] or different forms of electrical brain stimulation [7, 11, 12], which showed promising results for a positive effect on creativity.

Another prominent line of research investigates the effects of physical activity on creativity. Rominger et al. [13] observed a connection between everyday bodily movement and creative performance (for a meta-analysis see [14]). This raises the question of which kind of physical activity is most sensitive for variations in creative performance. A very basic approach in this context is to manipulate left- and right-hemispheric cortical activation levels through unilateral hand contractions [15–17]. The authors of these studies hypothesized, that by contracting the muscles of one arm, while the other arm is being hold at rest, sensorimotor cortex activation lateralized to the contralateral hemisphere causes a general increase of activation in that brain hemisphere. The explanation for these effects of unilateral muscle contractions on creative performance is based on the principle of spread of activation [18–21]. According to this principle, the activation of a certain brain area simultaneously leads to a rise in activation in other adjacent brain areas, which could be task-relevant. Goldstein et al. [15] found that left-hand contractions led to a significant increase, while right-hand contractions led to a tendential decrease in verbal creative performance, relative to a control group. Rominger et al. [16] extended the work of Goldstein et al. [15] by using a figural creativity task. Also, in contrast to Goldstein et al. [15], they used a pre-/posttest design in which all participants were tested on figural creativity before and after a hand contraction intervention. For the intervention they divided the participants into two groups, where one group performed left-hand contractions and the other group performed right-hand contractions. Similar to the effects found by Goldstein et al. [15], their results show increased performance on figural creativity tasks following left-hand contractions and a tendential inverse effect after right-hand contractions, relative to baseline. However, these results were moderated by positive schizotypy and only individuals scoring low on positive schizotypy showed the effects of hand contractions on creativity. Furthermore, there are also findings pointing in the opposite direction. Turner et al. [17] attempted a replication of the findings by Goldstein et al. [15]. In a very similar experiment the authors found that verbal creative performance was higher after right-hand contractions and lower after left-hand contractions in comparison to a control group, which is the exact opposite finding to those from Goldstein et al. [15].

Although the effects of unilateral hand-contractions on creativity as demonstrated in previous research [15–17] are intriguing, the underlying cortical mechanisms remain largely unclarified, also because cortical activation patterns have not been recorded during those experiments by means of neuroimaging techniques. Cross-Villasana et al. [22] present a possible explanation for this question. Their aim was to investigate specific processes taking place during and after unilateral hand contractions on a neuronal level. They let participants perform left- and right-hand contractions in two separate blocks, while simultaneously monitoring the cortical activity via EEG. Their results show a significant decrease of alpha power over regions of the sensorimotor cortex (SMC) *during* left- and right-hand contractions, compared

to the baseline. *After* contractions alpha power levels across the whole scalp were significantly higher compared to baseline but only in the left-hand condition. For the right-hand condition alpha amplitudes were close to those at baseline. This decrease in alpha power during hand contractions is also referred to as an event-related desynchronization (ERD) of the alpha band and is indicative of an increase of neuronal activity, while the subsequent increase of alpha power (after the termination of the hand contractions) is referred to as an event-related synchronization (ERS) of the alpha band, which is hypothesized to be cortical inhibition or, according to the traditional view, a reduced state of active information processing [23–25]. This phenomenon of a rebound in alpha power (i.e., alpha ERS) after the termination of a cognitive task (i.e., task-related alpha ERD) has also been described before in the context of neurofeedback training [26, 27]. Ros et al. [27] assume that the brain's self-regulating capabilities are responsible for these types of fluctuations in certain EEG frequency bands, such as alpha, to occur in order to maintain homeostasis of cortical activation. Considering that higher alpha power has been repeatedly associated with higher creative performance (see [2]), these results might explain the creativity enhancing effect of left-hand contractions. It can be assumed that an alpha ERD (decrease of alpha power) from active sensorimotor regions during the execution of a motor task, spreads to adjacent regions [18] and induces an alpha ERS (increase of alpha power) in the underlying cortical regions, after the termination of the motor task. This alpha ERS can be assumed to be the source of the beneficial effects of left-hand contractions on creative performance, as the results of Cross-Villasana et al. [22] show that widespread alpha ERS was present solely after left-hand but not right-hand contractions.

The specific aim of this study was to test whether the effects, identified by Goldstein et al. [15] and Rominger et al. [16], could be replicated by using other motor tasks to influence hemispherical asymmetry in the brain. Because hand contractions are simple movements, presumably leading to low levels of cortical activation in the hand and finger areas of the SMC, we assumed that a more advanced motor task should lead to higher levels of cortical activation and stronger effects on creative performance. For that reason, we chose to substitute the hand contraction task with a more motion-intensive task in the form of basketball dribbling, which we assume would lead to stronger SMC activation, because it mobilizes not only finger articulations but also the wrist, elbow, and, to a lesser extent, shoulder articulations. In extending previous studies, and to validate the influence that the motoric intervention has on the laterality of brain activation, we assessed the cortical activation during the execution of the dribbling task. For that we chose functional near-infrared spectroscopy (fNIRS).

FNIRS (also referred to as "NIRS") is an optical method, based on the principle of light absorption of hemoglobin in the cerebral blood flow [28]. Changes in the cerebral blood flow, specifically changes in the concentration of oxygenated hemoglobin (oxy-Hb) and deoxygenated hemoglobin (deoxy-Hb), can be registered to infer on cortical activation. The typical fNIRS signal is characterized by an increase in oxy-Hb and a simultaneous, albeit weaker decrease in deoxy-Hb after neural activity occurs in a cortical area. Similar to previous investigations, the general pattern of increased contralateral activation during motor execution has also been replicated with fNIRS, as summarized in a review by Leff et al. [29]. For instance, Franceschini et al. [30] found consistently higher contralateral than ipsilateral activation for voluntary hand movement, as well as for passive tactile and electrical hand stimulations. FNIRS has repeatedly shown to be a viable technique for the investigation of SMC activity, with the main benefit being its robustness against motion artifacts, which (under correct operation) is far superior to that of EEG or fMRI. Other advantages of fNIRS are its portability, relatively low cost, few contraindications, as well as the suitability for long-time monitoring and multimodal imaging alongside other neuroimaging methods like EEG or fMRI. Furthermore, fNIRS yields better spatial resolution than EEG and better temporal resolution than fMRI, as

well as the possibility to measure both oxy-Hb and deoxy-Hb simultaneously [31]. Carius et al. [32] already demonstrated the use of fNIRS to record sensorimotor activity simultaneously to basketball dribbling. Furthermore, our group previously investigated the signal quality of a subset of the present study's data and found that good contrast-to-noise ratio of the fNIRS signal was achieved during basketball dribbling [33].

In the present study half of the participants dribbled a basketball with the non-dominant left hand (NDH), the other half with the dominant right hand (DH), while simultaneously the hemodynamic activity was recorded over the SMC using fNIRS. It is hypothesized that hemodynamic activity patterns resemble those found in previous fNIRS studies investigating cortical activation during hand movements [29, 30]. Significantly higher contralateral than ipsilateral SMC activity, in relation to the side of the dribbling is expected. In addition, we were interested if we would be able to predict the intervention group membership of our participants (NDH vs. DH) through the SMC activation patterns. This procedure was inspired by a study by Reichert et al. [34] who used EEG-data to classify responders and non-responders of a neurofeedback paradigm. Good classification accuracy of the cortical activity patterns during a more complex motor task, such as basketball dribbling, could have implications for the design of future fNIRS neurofeedback and brain-computer-interface paradigms, which rely on accurate identification of brain activity patterns [35, 36]. Furthermore, the classification of intervention groups based on fNIRS measurements of brain activity could potentially be beneficial for the evaluation of new rehabilitation methods, e.g., for stroke [37, 38] or dysphagia [39], where the effects of different interventions on cortical activation are of primary interest. Concerning creative performance, verbal and figural divergent thinking (DT) tasks were presented to the participants before and after the dribbling intervention, in a pre-/posttest experimental design. Parallel to the results by Goldstein et al. [15] and Rominger et al. [16], it is hypothesized that NDH dribbling leads to an increase, whereas DH dribbling leads to a (tendential) decrease of verbal and figural DT performance in the posttest, relative to the pretest.

## Method

### Sample

In total 43 right-handed volunteers (23 men and 20 women) aged between 18 and 28 years ($M = 23.37$, $SD = 2.50$) participated in this study. The participants were pseudo-randomly allocated to either the DH ($n = 22$) or the NDH group ($n = 21$), with the only restriction being that groups had to be gender balanced. We additionally assessed the participants' experience in ball dribbling and their general sport-related habits as control variables, which were also balanced across the two intervention groups. The study was conducted at the University of Graz. Recruitment was done via e-mail distribution, flyer posting and social media. Prerequisites for participation were right-handedness (as assessed by the Edinburgh Handedness Inventory [40]) as well as the absence of neurological and psychiatric conditions. Psychology students received course credit for their participation. This study was conducted in accordance with the Declaration of Helsinki. Approval for the study procedures was given by the ethics committee of the University of Graz (GZ. 39/73/63 ex 2019/20).

### Material and tasks

#### Basketball dribbling intervention

Basketball dribbling was chosen to induce activation of the sensorimotor region. Participants had to dribble a standard (size 7) basketball 16 times for 10 sec each, while sitting on a chair. Every dribbling trial was followed by a randomized resting period of 10, 12, 13 or 15 sec,

where the participants held the ball still on their lap with both hands. A randomized inter-trial resting period was chosen to accommodate for slow cyclic blood pressure fluctuations (i.e., Mayer waves) that can have confounding effects in fNIRS data analysis [41, 42]. We requested each participant to dribble the ball either with their dominant right (DH) or non-dominant left (NDH) hand, depending on group assignment. Each participant should dribble the basketball in a way that they felt comfortable during each of the 16 ten-second-trials, while sitting upright on a chair with no wheels or armrests, ensuring the necessary stability and freedom of movement. They were not allowed to stop dribbling midway through a trial and were asked always to dribble the ball up to approximately the height of their waist. They were also allowed to look at the ball while dribbling to have better control over it. The signals to start and stop the dribbling were given by auditory cues, that were programmed into a script using Psy-choPy3 [43]. While dribbling with one hand, the participants should lay the other hand on their lap. The dribbling intervention was performed, while simultaneously the hemodynamic activity over the participants' SMC was recorded with fNIRS.

## Edinburgh handedness inventory

Handedness of our participants was assessed by means of a modified version of the Edinburgh Handedness Inventory (EHI; Oldfield [40]). The EHI includes questions about a person's hand preference for 10 different everyday activities (e.g., writing, drawing, etc.), for which the options "left", "right" or "both" hands are given, as well as an additional "Yes/No"-question for each item, whether the person ever uses the other hand. A laterality coefficient is calculated for the given responses, that ranges from -100 (fully left-handed) to +100 (fully right-handed), meaning high negative values are associated with left-handedness, values around 0 with ambidexterity and high positive values with right-handedness.

## Picture completion task (figural DT)

For the assessment of figural DT, a subtest of the Torrance Tests of Creative Thinking (TTCT; Torrance [44]), the Picture Completion Task (PCT) was used. The TTCT is a comprehensive test battery, that includes different creativity tasks, from which the PCT is a commonly used task in creativity research. The PCT that we presented in this study, was slightly modified from the original TTCT version. This PCT consisted of 10 items, which, by means of the odd-even-method, were split into two test forms (A & B) of five items each. The items consisted of a square box with an abstract figure in it. The participants had 1 minute for each item, to make a complete drawing out of the abstract figure, by adding lines to it. This drawing should be as creative as possible and incorporate the abstract figure in an original way. Then the participants should give the drawing an interesting title, for which they were given another 20 sec. For the scoring of the participants drawings we used a rating approach, which has proven to be a suitable method for scoring originality of DT task responses [45, 46]. Six raters (3 men and 3 women) aged between 26 and 35 years ($M = 27.83$, $SD = 3.55$) were instructed to score all PCT responses of the participants, on a five-point scale ranging from "1" (not original at all) to "5" (very original). Inter-rater reliability, as determined by the intraclass correlation coefficient (ICC = .86, 95% CI [.80, .91]), indicates good inter-rater agreement [47]. Figural DT of the participants for each time point equaled the average originality rating (given by the raters) of the participants' drawings.

## Regensburg word fluency test (verbal DT)

For the verbal DT component of the study, the Regensburg Word Fluency Test (original title: Regensburger Wortflüssigkeitstest; RWT; Aschenbrenner et al. [48]) was used. The RWT

includes five lexical and five semantic fluency tasks, as well as two lexical and two semantic fluency tasks with category-switching. In lexical tasks the aim is to name as many words as possible that start with a certain letter, whereas in semantic tasks the aim is to name as many examples for a certain category of things as possible. Switching-tasks include alternating between two given conditions (e.g., alternating between G- and R-words or alternating between sports and fruits). Time constraint for each task can be set to either 1 or 2 min. The authors give a recommendation for a selection of tasks for repeated testing with two measurement time points. These recommendations have been applied in this study to have two parallel forms (A & B; analogous to the PCT). RWT test form A consisted of the tasks "P-words", "G-/R-words", "Animals" and "Sports/Fruits". RWT test form B consisted of "M-words", "H-/T-words", "Food" and "Clothing/Flowers". For each individual task, the 2 min time limit was used, following the test authors' recommendations for the measurement of healthy adults. For the evaluation, participants' verbal responses were recorded. Scoring has been done according to the procedures described in the RWT manual. Verbal DT for each time point equaled the sum of raw scores of adequate responses per task, given by the participants.

### FNIRS recording and pre-processing

The participants' hemodynamic response was recorded during the dribbling intervention with a NIRSport2 (NIRx Medizintechnik GmbH, Berlin, Germany), which is a continuous-wave multichannel fNIRS device. The probe layout over the SMC consisted of eight dual-wavelength dual-tipped LED source- and eight active dual-tipped detector-optodes. The eight source-optodes of the fNIRS system emit light at 760 nm and 850 nm wavelengths, for the measurement of oxy-Hb and deoxy-Hb in the cortical blood flow. A sampling frequency of 10.2 Hz was used. Distance-holders were used to ensure a 30 mm distance between each source-detector-pair, which corresponded to long-separation channels. In addition, short-separation channels were implemented. To perform short-separation measurements, one detector-optode had to be used, to serve as a receiving optode for the short-separation detector probes (8 mm distance to source-optodes). Fig 1 provides a visualization of the probe that was used with all except for the first six participants (NDH = 4; DH = 2), for which a slightly different layout was used. This led to the issue that information from certain channels is missing in those participants (see Table 1). A sensitivity profile of the probe layout was calculated within Atlas-Viewer (Ver. 2.11.3; Aasted et al. [49]), a free software package suited for the use with the fNIRS data processing software Homer2 [50], based on MATLAB (MathWorks, Natick, MA, USA). The sensitivity profile gives an estimation of the cortical regions that were covered by our probe and was projected onto the Colin27 digital brain atlas [51] with default settings (1e6 photons, default optical properties; [49]).

The recorded fNIRS data was pre-processed using Homer2 (Ver. 2.3; Huppert et al. [50]). The following processing steps (function parameters in brackets) were conducted in this order: Raw intensity data was converted into optical density with the function *hmrIntensity2OD*. The *enPruneChannels* function was applied to exclude channels with low signal-to-noise ratio (dRange: -1e+04 / 1e+04, SNRthresh: 6.67, SDrange: 0.0 / 45.0, reset: 0). In the whole sample only four such channels were identified and excluded. All of them were short-separation channels, no long-separation channels were excluded. A wavelet filter was applied by using the function *hmrMotionCorrectWavelet* (iqr: 0.10). The functions *hmrMotionArtifact* (tMotion: 0.5, tMask: 1.0, STDEVthresh: 15.0, AMPthresh: 0.30) and *enStimRejection* (tRange: -10.0 / 20.0) were implemented to automatically identify and exclude trials which were still impacted by motion artifacts after the wavelet filter. The signal was bandpass filtered with *hmrBandpassFilt* (hpf: 0.01, lpf: 0.50) and converted into oxy-Hb and deoxy-Hb

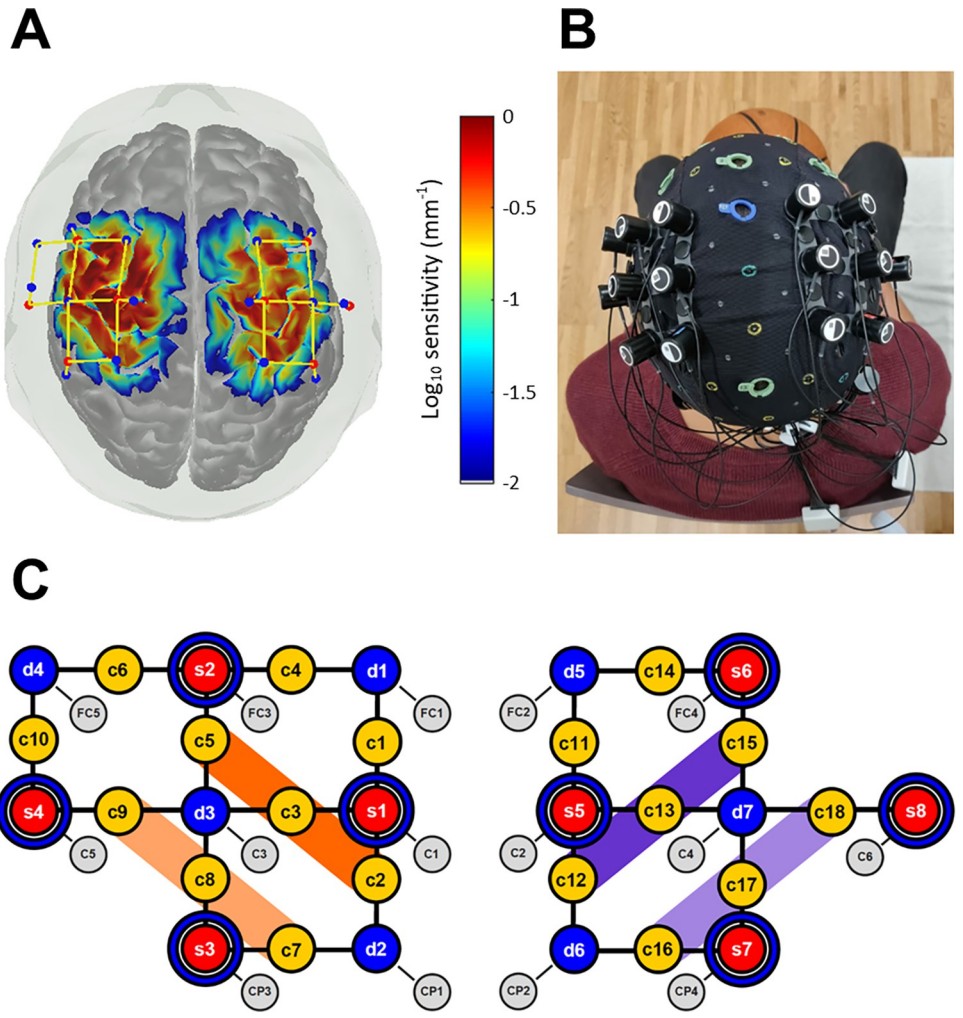

**Fig 1. Visualizations of the fNIRS probe.** (A) the sensitivity profile which shows that precentral as well as postcentral hand/arm regions were covered by the probe bilaterally, (B) the fNIRS source- and detector-bundles as they were applied on the head of a participant holding the basketball before the dribbling intervention, and (C) a diagram of the probe displaying the positions of the sources (s1–8, red circles), detectors (d1–7, blue circles; blue rings represent short-separation detectors), channels (c1–18, yellow circles; highlighting represents region of interest: orange = "Precentral Left", light-orange = "Postcentral Left", purple = "Precentral Right", light-purple = "Postcentral Right") as well as the approx. EEG 10–20 positions (grey circles) as reference points for the source and detector locations. (figures were partially reused and adapted from Kanatschnig et al. [33]).

concentrations with *hmrOD2Conc* (ppf: 6.0 / 6.0). In correspondence with NIRx support we implemented an adaptation to the *hmrOD2Conc* function, through which the optode-distances of all long-separation channels were fixed at 30 mm, corresponding to the 30 mm distance-holders used in our probe (see supporting information S1 File for the adapted Homer2 function script). General hemodynamic drift, as measured by the short-distance channels, was regressed out of the hemodynamic response function (HRF) by using *hmrDeconvHRF_DriftSS* (trange: -5.0 / 20.0, glmSolveMethod: 1, idxBasis: 2, paramsBasis: 0.1 / 3.0 / 10.0 / 1.8 / 3.0 / 10.0, rhoSD_ssThresh: 15.0, flagSSmethod: 0, driftOrder: 0, flag-MotionCorrect: 0). Block averages were calculated with *hmrBlockAvg* (trange: -5.0 / 20.0). For further statistical analysis, the task-related hemodynamic response was defined to be the average hemoglobin concentration in the time interval of seconds 5 to 12 after dribbling onset, which after visual inspection

of the HRF was found to be the time window in which peak activation was observable. The average hemodynamic response was exported channel-wise for all participants and for oxy-Hb and deoxy-Hb, respectively.

After motion artifact correction through the use of wavelet filtering only 11 of in total 688 experimental trials across all participants were excluded from the analysis. Wavelet filtering, which was first introduced for the use in fNIRS analysis by Molavi & Dumont [52], has shown to be a reliable method for the correction of motion artifacts in fNIRS data [53–55]. No manual correction method (i.e., exclusion of trials through visual inspection of the data) was performed. The maximum number of excluded trials from the recording of a single participant did not exceed two trials.

## Regions of interest

For statistical analysis, we divided the channels of our fNIRS probe into regions of interest (ROI) within the SMC. We defined the approximated locations of the cortical regions we measured with our probe within AtlasViewer by means of automated anatomical labeling (AAL; Tzourio-Mazoyer et al. [56]). Through the AAL method we obtained coordinates in Montreal Neurological Institute (MNI) space as well as labels of the underlying cortical regions for all channels. Based on the AAL labels we defined four ROIs that form symmetrical patterns across both brain hemispheres and comprise three channels each: "Precentral Left", "Postcentral Left", "Precentral Right" and "Postcentral Right". Therefore, our regions of interest can be subcategorized by the factors "Hemisphere" with the separation into left hemisphere (LH) and right hemisphere (RH), as well as "Sensorimotor Function" with the distinction between the precentral region (being responsible for motoric functions) and the postcentral region (being responsible for functions of sensation). All channel information is presented in Table 1.

**Table 1. Channel information of the fNIRS probe.**

| Ch. | Src. | Det. | Coord. MNI | Label AAL | ROI |
|-----|------|------|------------|-----------|-----|
| 1* | 1 [C1] | 1 [FC1] | -19–6 64 | Frontal_Sup_L | |
| 2 | 1 [C1] | 2 [CP1] | -17–24 66 | Precentral_L | Precentral Left |
| 3 | 1 [C1] | 3 [C3] | -29–15 62 | Precentral_L | Precentral Left |
| 4* | 2 [FC3] | 1 [FC1] | -27 7 58 | Frontal_Mid_L | |
| 5* | 2 [FC3] | 3 [C3] | -45 2 62 | Precentral_L | Precentral Left |
| 6* | 2 [FC3] | 4 [FC5] | -37 8 36 | Precentral_L | |
| 7 | 3 [CP3] | 2 [CP1] | -23–34 54 | Postcentral_L | Postcentral Left |
| 8 | 3 [CP3] | 3 [C3] | -39–24 55 | Postcentral_L | Postcentral Left |
| 9 | 4 [C5] | 3 [C3] | -55–14 56 | Postcentral_L | Postcentral Left |
| 10* | 4 [C5] | 4 [FC5] | -55–2 40 | Postcentral_L | |
| 11* | 5 [C2] | 5 [FC2] | 30 0 73 | Frontal_Sup_R | |
| 12 | 5 [C2] | 6 [CP2] | 28–26 70 | Precentral_R | Precentral Right |
| 13 | 5 [C2] | 7 [C4] | 42–14 73 | Precentral_R | Precentral Right |
| 14* | 6 [FC4] | 5 [FC2] | 27 6 49 | Frontal_Mid_R | |
| 15* | 6 [FC4] | 7 [C4] | 49–3 60 | Precentral_R | Precentral Right |
| 16 | 7 [CP4] | 6 [CP2] | 36–38 61 | Postcentral_R | Postcentral Right |
| 17 | 7 [CP4] | 7 [C4] | 53–27 65 | Postcentral_R | Postcentral Right |
| 18 | 8 [C6] | 7 [C4] | 44–16 42 | Postcentral_R | Postcentral Right |

"Ch.": channel number; "Src.": source number (approx. EEG position in brackets); "Det.": detector number (approx. EEG position in brackets); "Coord. MNI": Montreal Neurological Institute coordinates; "Label AAL": region label according to automated anatomical labeling; "ROI": Region of interest

*: data missing from first six participants.

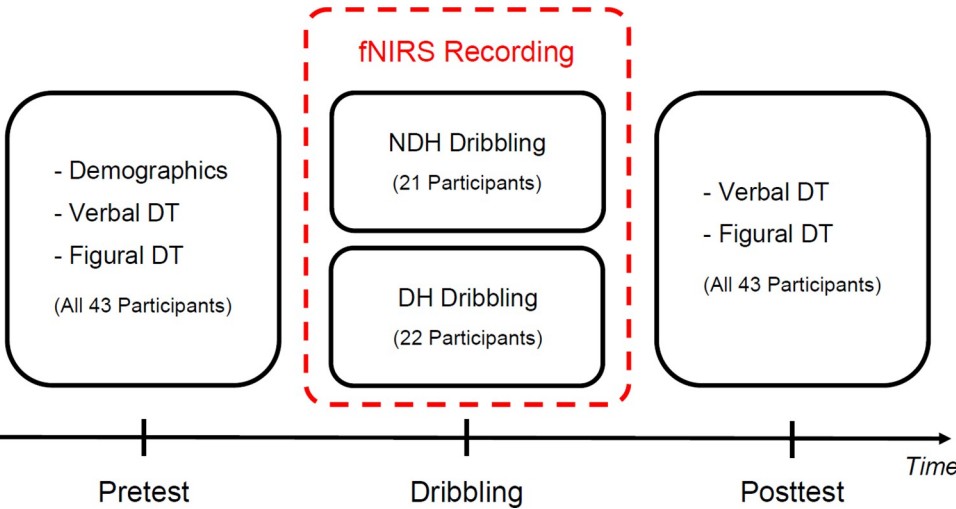

**Fig 2. Procedure.** Procedure of the present study ("DT": divergent thinking; "fNIRS": functional near-infrared spectroscopy; "NDH": non-dominant left hand; "DH": dominant right hand).

## Design and procedure

After giving written informed consent, the participants filled out a demographics questionnaire as well as the EHI. Two additional sport-related control variables were measured. Concerning dribbling, the participants were asked "Do you have experience with sports that involve dribbling a ball (e.g., basketball, handball, etc.)?" and concerning their general sport-related habits, the participants were asked "Do you currently do sports regularly (at least once a week)?" (~10 min). Then participants were instructed for the first part of DT tasks (pretest). First the RWT (~15 min) and then the PCT (~10 min) were administered. Then the fNIRS-cap, which had been prepared beforehand, was put onto the head of the participants (the Cz position was measured as reference for the positioning of the cap) and the fNIRS system was calibrated for measurement (~10 min). The participants were given the instructions for the dribbling intervention, as well as the possibility to make a few dribbles to get accustomed (~4 min). Then the dribbling intervention started. (~6 min). After dribbling, the second part of DT tasks (posttest), again starting with the RWT (~15 min) followed by the PCT (~10 min), was administered. Test form A consisted of one half, test form B consisted of the other half of RWT and PCT items. Presentation of versions A and B, as pretest or posttest, was pseudorandomized and equal distribution across intervention-groups and gender was guaranteed. Total testing time was 90 min. Fig 2 summarizes the study design and the procedure.

## Statistical analysis

To test whether the participants were equally distributed chi-square tests of independence with continuity correction were calculated with dribbling group and gender, test order, dribbling experience and sport habits, respectively. T-tests were calculated to compare mean age and EHI scores between groups.

For the analysis of intervention related changes from the pretest to the posttest, two 2x2 ANOVAs for repeated measures were calculated for verbal and figural DT, respectively. The ANOVAs included the between-subjects factor "Dribbling Group" (DH vs. NDH) and the within-subject factor "Time" (Pretest vs. Posttest). One participant of the DH group had to be excluded from the analysis of verbal DT, because it was discovered midway through the RWT

measurement, that there had been a misunderstanding regarding the instructions. A significant interaction effect of "Dribbling Group * Time" would indicate a significant effect of the dribbling intervention, depending on which hand was used.

To evaluate our hypotheses regarding cortical hemodynamic activity, we calculated two 2x2x2 ANOVAs with the between-subjects factor "Dribbling Group" (DH vs. NDH), as well as the within-subjects factors "Hemisphere" (LH vs. RH) and "Sensorimotor Function" (Motor vs. Sensation). The dependent variables for each ANOVA were the task-related concentration changes in oxy-Hb and deoxy-Hb, respectively. These ANOVAs only included data from the channels previously defined in the ROIs (see Table 1). The oxy-Hb and deoxy-Hb concentration values were averaged across all channels of the respective ROI. For the prediction of group membership, the averaged hemodynamic response variables from all ROI and oxy-Hb as well as deoxy-Hb (eight variables in total) were entered into a linear discriminant analysis (LDA) as predictors for "Dribbling Group" membership (DH vs. NDH). All statistical calculations were performed using IBM SPSS Statistics 29 (IBM Corp., Armonk, NY, USA). The significance level for all analyses was set to 0.05 (two-tailed). For all necessary post-hoc comparisons Bonferroni-adjusted post-tests were performed (see supporting information S2 File for a SPSS data file including all relevant variables and S3 File for a SPSS syntax file including all calculations of the main statistical analysis).

## Results

Table 2 shows descriptive and comparative statistics for sociodemographic and control variables. Neither the chi-square nor t-tests yielded a significant result (all $p \geq .702$). Therefore, it can be assumed that the intervention groups were well balanced regarding the examined variables.

### Sensorimotor cortex activation

Hemoglobin concentrations are presented in units of micromolar (μM). For oxy-Hb, the analysis yielded a significant interaction effect for "Hemisphere * Dribbling Group" ($F(1, 41) = 6.17$, $p = .017$, $\eta_p^2 = .13$). Post-tests indicate that the NDH group ($M = 0.24$, $SE = 0.03$) had a significantly stronger oxy-Hb increase in the RH compared to the DH group ($M = 0.11$, $SE = 0.03$; $p = .005$). The two groups did not differ in oxy-Hb levels in the LH ($p = .941$). Furthermore, the DH group showed a significantly stronger oxy-Hb increase in the LH ($M = 0.20$, $SE = 0.04$) compared to the RH ($M = 0.11$, $SE = 0.03$; $p = .022$); whereas the NDH group

**Table 2. Descriptive and comparative statistics for sociodemographic and control variables.**

| | Dribbling group: | | Statistics: | |
|---|---|---|---|---|
| | NDH ($n = 21$) | DH ($n = 22$) | $\chi^2$ (df = 1) | $p$ |
| **Gender (male/female)** | 11/10 | 12/10 | < 0.01 | > .999 |
| **Test order (AB/BA)** | 10/11 | 11/11 | < 0.01 | > .999 |
| **"Do you have experience with sports that involve dribbling a ball (e.g., basketball, handball, etc.)?" (Yes/No)** | 12/9 | 11/11 | 0.03 | .870 |
| **"Do you currently do sports regularly (at least once a week)?" (Yes/No)** | 18/3 | 19/3 | < 0.01 | > .999 |
| | | | $t$ (df = 41) | $p$ |
| **Age** | 23.52 (±2.56) | 23.23 (±2.49) | 0.39 | .702 |
| **EHI** | 80.24 (±15.04) | 80.23 (±15.23) | < 0.01 | .998 |

For categorical variables absolute participant counts are presented for the respective categories. For continuous variables group means are presented, with standard deviations in parentheses. Comparative statistics results are provided in the "Statistics" column. "NDH": non-dominant left hand; "DH": dominant right hand"; "EHI": Edinburgh Handedness Inventory laterality coefficient, high positive values indicate right-handedness.

showed no significant hemispheric difference ($p$ = .262). No other main or interaction effects were significant ["Dribbling Group": $F$ (1, 41) = 1.78, $p$ = .190, $\eta_p^2$ = .04; "Hemisphere": $F$ (1, 41) = 0.73, $p$ = .397, $\eta_p^2$ = .02; "Sensorimotor Function": $F$ (1, 41) = 0.85, $p$ = .361, $\eta_p^2$ = .02; "Sensorimotor Function * Dribbling Group": $F$ (1, 41) = 2.96, $p$ = .093, $\eta_p^2$ = .07; "Hemisphere * Sensorimotor Function": $F$ (1, 41) = 1.09, $p$ = .303, $\eta_p^2$ = .03; "Hemisphere * Sensorimotor Function * Dribbling Group": $F$ (1, 41) = 0.29, $p$ = .591, $\eta_p^2$ = .01)].

In the case of deoxy-Hb, the main effect "Sensorimotor Function" reached significance ($F$ (1, 41) = 13.77, $p$ < .001, $\eta_p^2$ = .25). This indicates that there was a significantly stronger deoxy-Hb decrease in the precentral region ($M$ = -0.06, $SE$ = 0.01) compared to the postcentral region ($M$ = -0.04, $SE$ = 0.01) across both hemispheres (note that in the case of deoxy-Hb lower values indicate higher cortical activation). The interaction effect for "Hemisphere * Dribbling Group" was significant ($F$ (1, 41) = 4.09, $p$ = .0497, $\eta_p^2$ = .09). Post-tests indicate that the DH group had a significantly stronger deoxy-Hb decrease in the LH ($M$ = -0.06, $SE$ = 0.02) compared to the RH ($M$ = -0.03, $SE$ = 0.02, $p$ = .018). In the NDH group there was no hemispheric difference ($p$ = .678); moreover, the two groups did not differ in deoxy-Hb levels in the LH ($p$ = .932) or the RH ($p$ = .141). No other main or interaction effects became significant ["Dribbling Group": $F$ (1, 41) = 0.49, $p$ = .486, $\eta_p^2$ = .01; "Hemisphere": $F$ (1, 41) = 2.03, $p$ = .162, $\eta_p^2$ = .05; "Sensorimotor Function * Dribbling Group": $F$ (1, 41) = 1.15, $p$ = .289, $\eta_p^2$ = .03; "Hemisphere * Sensorimotor Function": $F$ (1, 41) = 2.15, $p$ = .151, $\eta_p^2$ = .05; "Hemisphere * Sensorimotor Function * Dribbling Group": $F$ (1, 41) = 0.37, $p$ = .549, $\eta_p^2$ = .01)]. Fig 3 visualizes the interaction effect "Hemisphere * Dribbling Group" in oxy-Hb as well as deoxy-Hb. Figs 4 and 5 show the time series of the hemodynamic response functions averaged across each ROI for the NDH and the DH group, respectively.

For the exploratory analysis regarding the classification of group membership of the NDH and DH dribbling groups, a LDA with the hemoglobin variables from all ROI ("Precentral Left", "Postcentral Left", "Precentral Right", "Postcentral Right") and for oxy-Hb and deoxy-Hb, respectively (resulting in a total of eight variables) as predictors was performed. The overall results of the LDA indicate that the discriminant function is able to separate between the two intervention groups significantly better than if classification were performed randomly (Wilks' $\Lambda$ = .389; $\chi^2$ (8) = 34.95, $p$ < .001). The cross-validated classification results of the LDA

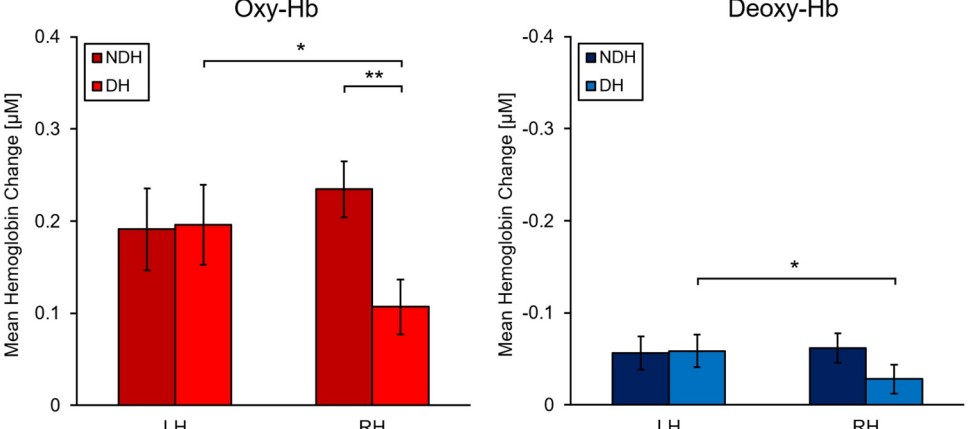

**Fig 3. ANOVA results.** A comparison of mean oxy-Hb and deoxy-Hb concentration change in the sensorimotor cortex during basketball dribbling. In the case of deoxy-Hb higher cortical activation is indicated by a decrease of hemoglobin (note the inversed scaling of the y-Axis). Error bars indicate standard errors ("NDH": non-dominant left hand [dark red/dark blue]; "DH": dominant right hand [red/blue]; "LH": left brain hemisphere; "RH": right brain hemisphere; *: $p$ < .05; **: $p$ < .01).

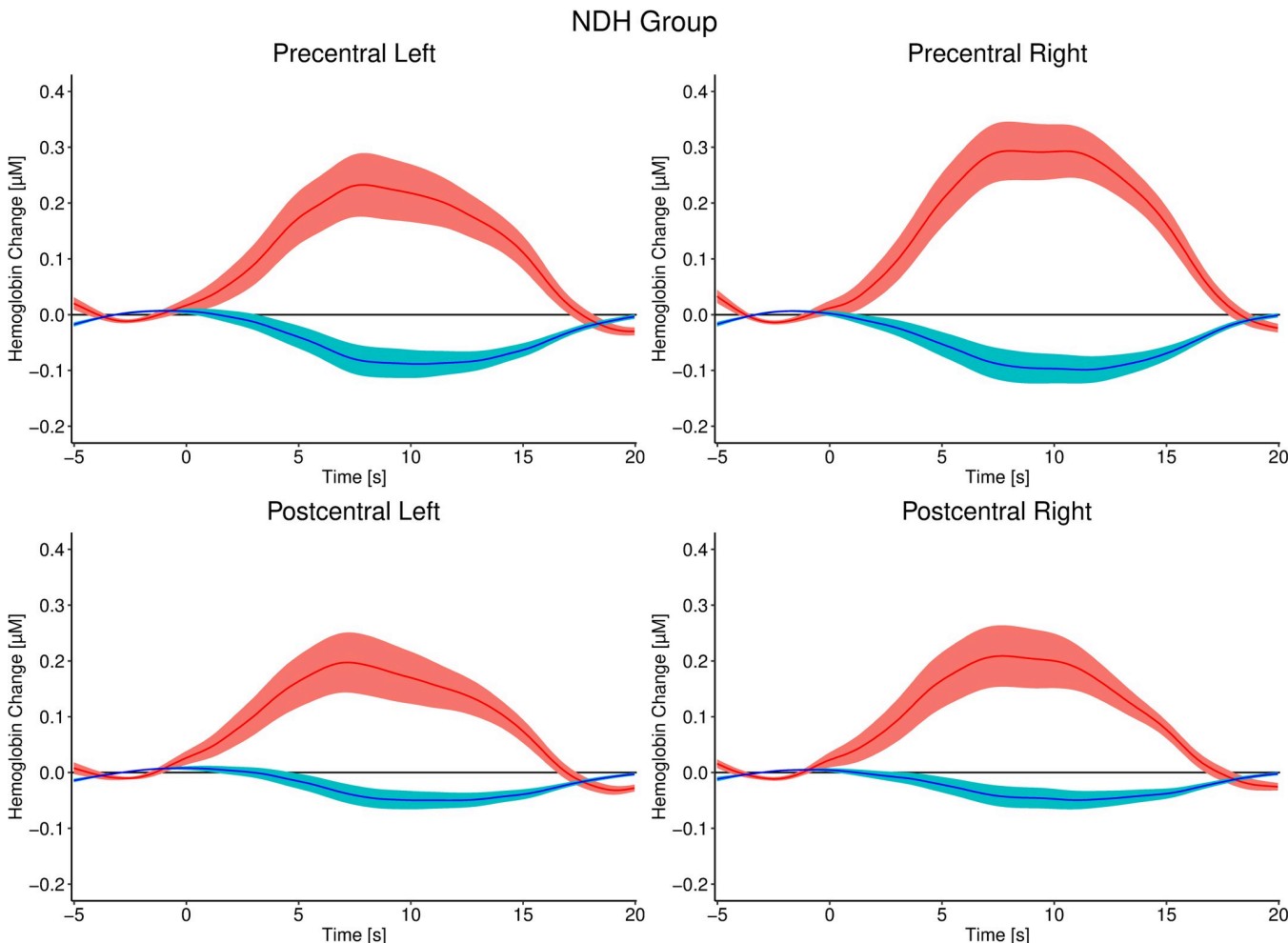

**Fig 4. A comparison of the HRF between ROIs in the NDH group.** Onset of basketball dribbling was at sec 0, offset at sec 10. Red graphs represent the change of oxy-Hb, blue graphs the change in deoxy-Hb per ROI in units of micromolar (μM). The light red and light blue areas around the graphs represent standard errors of oxy-Hb and deoxy-Hb, respectively. Statistical analyses were performed on concentration averages of the HRFs per ROI (sec 5–12; "NDH": non-dominant left hand).

show an accuracy of 86.0%. Accordingly, 18 of 21 participants in the NDH group and 19 of 22 participants in the DH group were correctly assigned to their respective groups based on their sensorimotor activation patterns (see Table 3).

## Creative performance

The analysis of verbal DT performance yielded a significant main effect of "Time", indicating higher performance in the posttest ($M = 114.45$, $SE = 2.59$) compared to the pretest ($M = 108.14$, $SE = 2.41$; $F(1, 40) = 9.72$, $p = .003$, $\eta_p^2 = .20$). Neither the main effect "Dribbling Group" ($F(1, 40) = 1.15$, $p = .291$, $\eta_p^2 = 03$), nor the interaction effect "Time * Dribbling Group" ($F(1, 40) = 1.36$, $p = .251$, $\eta_p^2 = .03$) reached significance. The analysis of figural DT performance revealed no significant main effects of "Time" ($F(1, 41) = 0.10$, $p = .758$, $\eta_p^2 < .01$) or "Dribbling Group" ($F(1, 41) = 0.04$, $p = .842$, $\eta_p^2 < .01$). Also, the interaction effect "Time * Dribbling Group" was not significant ($F(1, 41) = 0.01$, $p = .912$, $\eta_p^2 < .01$). Descriptive statistics of DT performance are presented in Table 4.

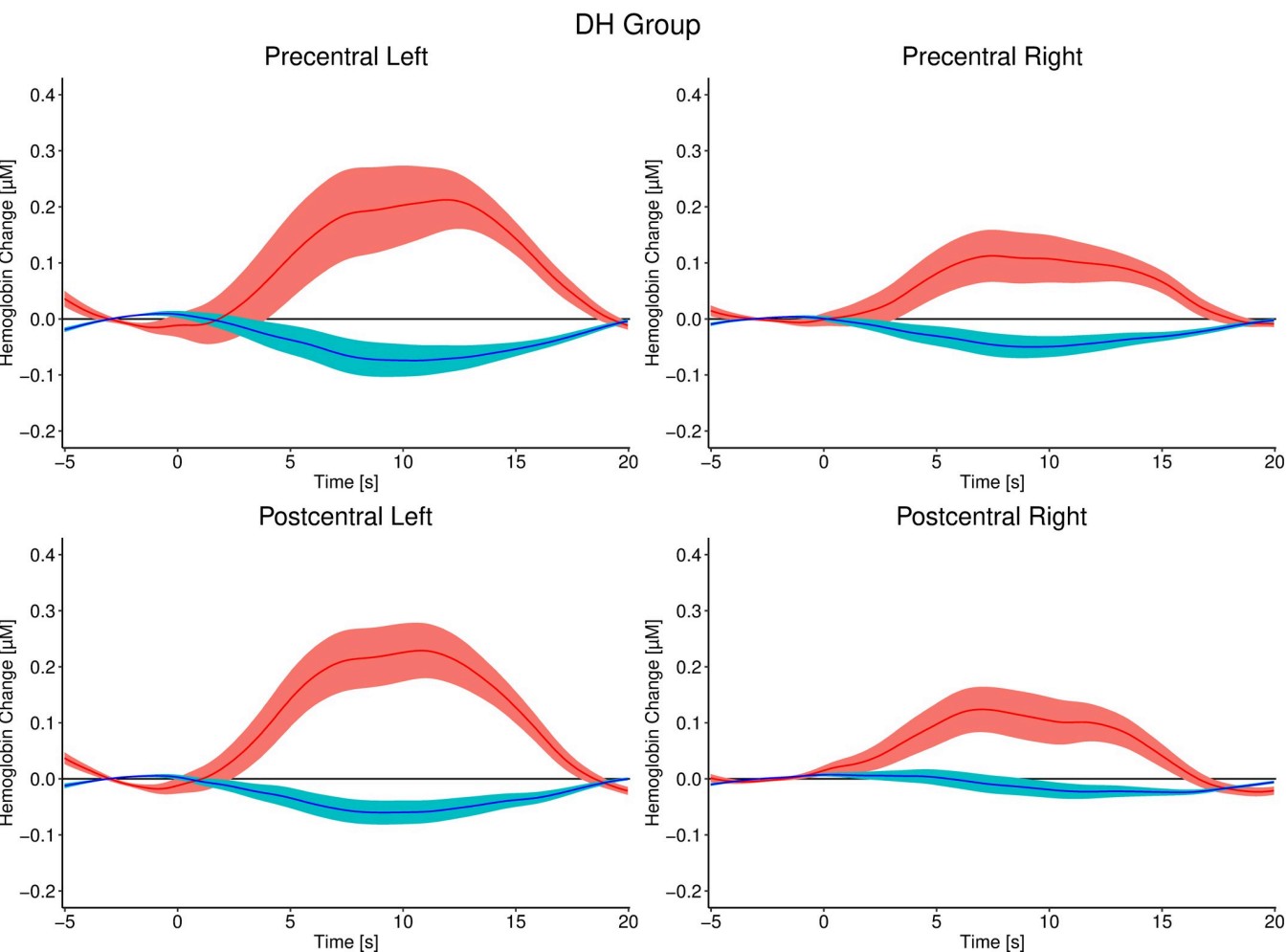

**Fig 5. A comparison of the HRF between ROIs in the DH group.** Onset of basketball dribbling was at sec 0, offset at sec 10. Red graphs represent the change of oxy-Hb, blue graphs the change in deoxy-Hb per ROI in units of micromolar (μM). The light red and light blue areas around the graphs represent standard errors of oxy-Hb and deoxy-Hb, respectively. Statistical analyses were performed on concentration averages of the HRFs per ROI (sec 5–12; "DH": dominant right hand).

## Discussion

The aim of this study was to investigate the neural underpinnings of left- and right-hand basketball dribbling and its relation to creative performance. Previous studies in the field of creativity enhancement suggest that performance on divergent thinking tasks can be enhanced through left-hand contractions [15, 16]. However, a contrary pattern of findings has also been reported in previous studies (i.e., higher performance through right hand contractions [17]). We aimed at replicating the originally found patterns of increased creative performance after left-hand contractions with a more advanced motor task in the form of basketball dribbling, to potentially induce a stronger effect of performance improvement. We tested 43 participants on verbal as well as figural divergent thinking tasks before and after a basketball dribbling intervention (dominant right-hand dribbling, DH, $n$ = 22; non-dominant left-hand dribbling, NDH, $n$ = 21) in this study. To investigate the expected asymmetry in sensorimotor activation depending on the hand used for dribbling, we used fNIRS to record cortical activity during the basketball dribbling intervention. Our results failed to show the expected effects of basketball

**Table 3. Classification results of the linear discriminant analysis for group membership.**

|  |  | Predicted group membership | | |
|---|---|---|---|---|
|  |  | NDH | DH | Total |
| **Original count** [a] | **NDH** | 19 | 2 | 21 |
|  | **DH** | 3 | 19 | 22 |
| **%** | **NDH** | 90.48 | 9.52 | 100 |
|  | **DH** | 13.64 | 86.36 | 100 |
| **Cross-validated count** [b] | **NDH** | 18 | 3 | 21 |
|  | **DH** | 3 | 19 | 22 |
| **%** | **NDH** | 85.71 | 14.29 | 100 |
|  | **DH** | 13.64 | 86.36 | 100 |

a. 88.4% of original grouped cases correctly classified.

b. 86.0% of cross-validated grouped cases correctly classified. In cross validation, each case is classified by the functions derived from all cases other than that case.

"NDH": non-dominant left hand; "DH": dominant right hand.

dribbling on creative performance. However, the analysis of cortical hemodynamic activity in the SMC revealed patterns of stronger bilateral cortical activation in the NDH group compared to the DH group. It is possible that these sensorimotor activity patterns happened due to the motoric complexity of the dribbling intervention. In the following, our results will be discussed in detail.

## Sensorimotor hemodynamic activity during basketball dribbling

As expected, the hemodynamic response, indicated by significant changes in oxy-Hb and deoxy-Hb concentrations in the cortical blood flow, was significantly impacted by the basketball dribbling intervention. The hypotheses concerning the underlying cortical activity in the SMC were partially confirmed. Overall, the DH group (dribbling with the dominant right hand) showed significantly higher contralateral compared to ipsilateral SMC activation both in oxy-Hb as well as deoxy-Hb. The NDH group (dribbling with the non-dominant left hand) also showed tendentially higher contralateral compared to ipsilateral activation in oxy-Hb and deoxy-Hb, however these hemispheric differences were not significant. The ipsilateral activation of the NDH group, was comparably high as the contralateral activation of the DH group, as indicated by oxy-Hb and deoxy-Hb concentrations showing no significant difference in the

**Table 4. Descriptive statistics of divergent thinking performance.**

|  | NDH ($n$ = 21) | DH ($n$ = 22) | Total ($n$ = 43) |
|---|---|---|---|
| **Verbal DT performance (RWT):** |  |  |  |
| **Pretest** | 111.76 (±2.94) | 104.52 (±3.71) | 108.14 (±2.41) |
| **Posttest** | 115.71 (±3.44) | 113.19 (±3.94) | 114.45 (±2.59) |
| **Figural DT performance (PCT):** |  |  |  |
| **Pretest** | 2.42 (±0.09) | 2.38 (±0.09) | 2.40 (±0.06) |
| **Posttest** | 2.38 (±0.12) | 2.37 (±0.10) | 2.38 (±0.08) |

Means and standard errors (in parentheses) of pretest and posttest measurements are presented for verbal DT (RWT sum score of correct responses) as well as figural DT (average originality rating), respectively. "DT": divergent thinking; "RWT": Regensburg Word Fluency Test; "PCT": Picture Completion Task; "NDH": non-dominant left hand; "DH": dominant right hand".

left SMC across the two groups. Furthermore, the NDH group showed significantly higher activation in the RH than the DH group but only in oxy-Hb. In terms of the generally expected patterns of higher contralateral compared to ipsilateral activation, our results are in line with the literature on motoric activation measured with fNIRS [29, 30]. However, the NDH group did not show the expected effect of significantly higher contralateral, relative to ipsilateral activation. Regarding the role of the factor "Sensorimotor Function", only the analysis of deoxy-Hb yielded a significant result, signifying higher hemodynamic activity in motor (precentral) regions compared to sensation (postcentral) regions bilaterally. The precentral region showing stronger activation in the motoric intervention is a very plausible result and it is in line with the notion that deoxy-Hb is a more reliable spatial indicator for cortical activation in the fNIRS signal [29].

The fact that there was no significant hemispheric difference in activation in the NDH group, reflects the comparably high spread of activation in the LH and the RH, meaning that there was no significant shift in activation towards the RH in the NDH group. It was shown in previous studies, that higher complexity of a motor task [32, 57–59] results in higher bilateral cortical activation intensity. Carius et al. [32] showed that different levels of task complexity of a basketball dribbling task induce altering hemodynamic response patterns, and that a higher level of perceived difficulty is associated with higher hemodynamic response in ipsilateral sensorimotor regions. The results of Verstynen et al. [57] and Haaland et al. [58] reveal that ipsilateral SMC activation during left-hand movement is stronger during the execution of a more complex motor task, relative to a more simple motor task and Alahmadi et al. [59] found that higher complexity of a hand-gripping task resulted in stronger involvement of the left hemisphere. The difficulty of the dribbling task, which presumably varied from participant to participant based on their skill level, could have played a role in the present study as well. It can be argued, although the dribbling intervention was designed to be as simple as possible, that the use of the non-dominant left hand in the NDH group could have made the task more difficult for those participants.

Moreover, the results of a fNIRS study by Lee et al. [60] show very similar results to our experiment. In that study, the authors let participants perform a complex motor task with the left or right hand involving the use of chopsticks. Analogous to our results, the use of the right hand let to a contralateral whereas the use of the left hand let to a more bilateral activation pattern of the fNIRS signal. A potential explanation for these patterns can be found in the role of the corpus callosum and theories of hemispheric inhibition and excitation [61, 62]. According to these theories, cortical activation in one hemisphere leads to a concurrent inhibition or excitation of activity in the corresponding cortical area in the other hemisphere. Due to our participants being exclusively right-handed and according to the theory of inhibition, potentially the RH in the NDH group was less successful in inhibiting the activity of the LH, resulting in the more bilateral activation pattern. However, according to the theory of excitation bilateral activation can also be seen as a recruitment of contralateral regions in order to provide additional resources for the processing of a more complex task. As van der Knaap & van der Ham [61] conclude, it remains difficult to pinpoint what specific processes are involved during inter-hemispheric interactions, albeit a combination of inhibitory as well as excitatory processes seems likely.

Our exploratory analysis of dribbling group classification by means of the LDA showed that rather high classification accuracy can be achieved by using fNIRS SMC activity data as classification predictors. 86.0% of the participants were correctly classified to their respective groups of NDH and DH dribbling after cross-validation. This is an interesting result, as it shows that good classification accuracy can also be achieved during more complex motoric interventions. Moreover, this has implications for future endeavors in the development of real-time fNIRS data classification systems such as neurofeedback paradigms or brain-computer-interfaces [35,

36]. In that regard, the field of motor rehabilitation is especially interesting. Kober et al. [63] demonstrated the effect that real fNIRS neurofeedback has compared to sham feedback during a hand motor imagery task and in a recent effort Hramov et al. [64] developed a classification method that can be used to classify both real as well as imaginary hand movements. As in these examples, motor execution and imagery paradigms are often based on simple hand movements. However, as our results and previous findings on more complex motor tasks show, SMC activation becomes increasingly more bilateral for left-sided movement, whereas SMC activation during right-sided movement remains more pronounced on the contralateral side of the brain. This will have to be considered when, through the robustness of fNIRS, motoric rehabilitation paradigms become more and more flexible and suited for increasingly advanced real-life motor tasks.

### Effects of basketball dribbling on creativity

Additionally, this study aimed at replicating findings from the research field of creativity enhancement [15, 16]. Our results show that overall, NDH basketball dribbling did not increase creative performance and DH dribbling did not decrease creative performance. Hence, the results of Goldstein et al. [15] and Rominger et al. [16] were not replicated by using basketball dribbling. Yet, the statistical analysis yielded a significant effect for the participants' verbal DT performance, namely that higher performance was observed in the posttest compared to the pretest across both intervention groups. It is likely however, that the higher performance in the posttest simply resulted from a training effect. In terms of figural creativity, the statistical analysis showed no significant differences in performance from pretest to posttest across both groups.

This raises the question why our results are different from those of the previous studies. First, we did not control for any further personality traits. Such traits could have helped to further explain the underpinnings of cognitive effects of unilateral movements on creativity, as it was the case with positive schizotypy in the study of Rominger et al. [16]. Second, the RWT, which we used in this study to measure verbal DT performance, did not include a "be creative" instruction. Although participants were instructed to name as many *different* items they can think of (i.e., using a divergent thinking style), the lack of a concrete instruction to give creative answers potentially lowered the influence of creative cognition during the task. Third, Goldstein et al. [15] applied a different task, namely the Remote Associates Test (RAT), to assess the changes in creative performance. Although considered to be a test of *divergent thinking* by Goldstein et al. [15], the RAT has been associated with a *convergent thinking* style by other research groups (e.g., [12, 65, 66]). The RWT however, is considered to be a test for verbal DT [48]. Given the fact that we wanted to examine verbal and figural DT performance, the RWT is arguably a better fit for this task compared to the RAT. Nevertheless, these differences in methodology might in part explain the differences in the reported effects. This would need further investigation. Another important factor to be considered is the basketball dribbling intervention. This has been hypothesized to be the underlying factor for the temporary enhancement of creative performance [15–17]. We expected significantly higher contralateral compared to ipsilateral SMC activation for both NDH and DH dribbling. What we found however, was that NDH dribbling lead to a more bilateral activation pattern, where the SMC activity on the contralateral (right) hemisphere was not significantly stronger than in the ipsilateral (left) hemisphere. This could mean that spread of activation took place across both brain hemispheres to a comparable level, implying that no strong lateralization of activity took place.

At this point of argumentation it should be noted that during the recent years empirical evidence increased [67, 68], which indicates the absence of beneficial effects of unilateral

movements on various behaviors (e.g., response times). Therefore, the present study adds to this line of null findings questioning a strong effect of unilateral movements on behavior. To summarize, the state of knowledge on creativity enhancement through the lateralization of cortical activity remains inconclusive. Furthermore, considering the results by Turner et al. [17]–who could not replicate the findings of Goldstein et al. [15] either and found an exact opposite results pattern instead–the underlying mechanisms of unilateral body movement and its effects on higher cognitive processes appear to be more complex.

## Limitations

A limitation of this and previous studies is that there are no clear indications yet on how long the effects of unilateral movements last. In this study we used verbal (~15 min) as well figural DT (~10 min) tasks, resulting in a combined 25 min of creativity testing both before and directly after the dribbling intervention. It is unclear whether the effects of the dribbling intervention lasted long enough to have an impact on both DT measurements in the posttest of our experiment. It is possible that the figural DT task, which came after the verbal DT task, was not influenced by the dribbling intervention if the effect of the intervention had already faded out at that point. This could potentially explain the lack of any significant effects in the analysis of figural DT performance. Thus, the order of task presentation could also have had an impact on the results. In this study the verbal assessment always came before the figural assessment. If the task order would have been interchanged or randomized, the results might have been quite different. Therefore, it may be advisable for future studies to use shorter tasks to measure the effects on a cognitive process of interest and also to control for potential task order effects.

Due to the motion-intensive task (i.e., basketball dribbling) in our study we used fNIRS for the measurement of cortical activation. This meant that we would gain the robustness against motion artifacts that the fNIRS technology holds. However, it also meant that we would have a detriment in the temporal resolution of the cortical signal, due to the slow nature of the fNIRS signal. Considering that a possible factor linking motor activation to creative performance being the rebound in alpha power after movement termination [22, 26, 27], future investigations into this matter should consider using EEG for the investigation of cortical activity, due to its superior temporal resolution over fNIRS. In the present study the recording of cortical activity was limited to solely the dribbling intervention and the SMC. Hence, it should also be considered to not only record cortical activity during the intervention, but also to record the cortical activity during subsequent behavioral tasks and over a larger area of the cortical surface. This would also give further insight into the underlying mechanisms of spread of activation in the brain. Such a procedure could help to better understand the effects of activation priming interventions on cognitive processes, and beyond that, better clarify how long such effects last.

Another aspect that must be considered is that we did not specifically control for the dribbling frequency, i.e., the number of times a participant dribbled the ball during each 10 second trial. Although all participants showed quite similar dribbling patterns–as they were all instructed to continuously dribble the basketball at a relaxed pace, to be able to keep control of it–an accurate assessment of the dribbling frequency would have been beneficial to control for potential influences on the hemodynamic response and creative performance.

## Conclusions

Overall, the analysis of sensorimotor hemodynamic activation, evoked by the basketball dribbling intervention, revealed a pattern that largely resembles the findings from previous investigations of sensorimotor activation during complex motor tasks. We found that right

dominant hand dribbling led to stronger contralateral activation, whereas left non-dominant hand dribbling led to a more bilateral activation pattern. Our results show how basketball dribbling affects SMC activity and that significant hemispherical differences are observable, depending on which hand is used for dribbling. Moreover, we were able to show that high intervention group classification could be achieved using fNIRS data. However, the hypothesized effects on creative performance were not confirmed in this study. The statistical analyses showed no differentiating effect on creative performance for the use of the left or right hand in the dribbling intervention. The present study once more illustrates that fNIRS is a viable technique for the investigation of cortical activation during physical activity and has implications for future endeavors in the development of motor rehabilitation paradigms.

## Supporting information

**S1 File. Homer2 script.** Modified Homer2 "hmrOD2Conc" function script.
(M)

**S2 File. SPSS data.** SPSS data file including all relevant variables for the statistical analysis.
(SAV)

**S3 File. SPSS syntax.** SPSS syntax file including all calculations of the main statistical analysis.
(SPS)

## Acknowledgments

We would like to thank all participants and raters for their contribution to this study as well as NIRx technical support for their help with the modification of the fNIRS analysis script.

## Author Contributions

**Conceptualization:** Thomas Kanatschnig, Andreas Fink, Silvia Erika Kober.

**Data curation:** Thomas Kanatschnig.

**Formal analysis:** Thomas Kanatschnig.

**Investigation:** Thomas Kanatschnig, Silvia Erika Kober.

**Methodology:** Thomas Kanatschnig, Christian Rominger, Andreas Fink, Silvia Erika Kober.

**Project administration:** Thomas Kanatschnig, Silvia Erika Kober.

**Resources:** Christian Rominger, Andreas Fink, Guilherme Wood, Silvia Erika Kober.

**Supervision:** Andreas Fink, Guilherme Wood, Silvia Erika Kober.

**Validation:** Christian Rominger, Andreas Fink, Guilherme Wood, Silvia Erika Kober.

**Visualization:** Thomas Kanatschnig.

**Writing – original draft:** Thomas Kanatschnig.

**Writing – review & editing:** Thomas Kanatschnig, Christian Rominger, Andreas Fink, Guilherme Wood, Silvia Erika Kober.

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
