## [Decision Letter · Decision Letter 0]

18 Jan 2023

PONE-D-22-32438Sensorimotor cortex activity during basketball dribbling and its relation to creativityPLOS ONE

Dear Dr. Kanatschnig,

Thank you for submitting your manuscript to PLOS ONE. After careful consideration, we feel that it has merit but does not fully meet PLOS ONE’s publication criteria as it currently stands. Therefore, we invite you to submit a revised version of the manuscript that addresses the points raised during the review process.

We look forward to receiving your revised manuscript.

Kind regards,

Vanessa Carels

Staff Editor

PLOS ONE

Journal Requirements:

We would like to thank all participants and raters for their contribution to this study as well as NIRx technical support for their help with the modification of the NIRS analysis script. The authors acknowledge the financial support by the University of Graz."

"The authors received no specific funding for this work. All authors are employees of the University of Graz, Austria."

Reviewers' comments:

Reviewer's Responses to Questions

**Comments to the Author**

1. Is the manuscript technically sound, and do the data support the conclusions?

Reviewer #1: Yes

Reviewer #2: Yes

Reviewer #3: Yes

2. Has the statistical analysis been performed appropriately and rigorously? 

Reviewer #1: Yes

Reviewer #2: Yes

Reviewer #3: Yes

3. Have the authors made all data underlying the findings in their manuscript fully available?

Reviewer #1: Yes

Reviewer #2: Yes

Reviewer #3: Yes

4. Is the manuscript presented in an intelligible fashion and written in standard English?

Reviewer #1: Yes

Reviewer #2: Yes

Reviewer #3: Yes

5. Review Comments to the Author

Reviewer #1: 03-01-2023

Review of PONE-D-22-32438

Sensorimotor cortex activity during basketball dribbling and its relation to creativity

Thank you for the opportunity to review this manuscript. The current article is an original research article on sensorimotor cortex activity during basketball dribbling and its relation to creativity. For that purpose, the authors applied functional near-infrared spectroscopy (fNIRS) as an established method to measure hemodynamic changes in selected brain regions. The aim of this study was to replicate previous findings on enhanced creative performance by increasing the level of activity in one of the brains hemispheres through unilateral hand movements. The novel aspects of this study can be found in the incorporation of basketball dribbling as a more advanced motor task and the application of fNIRS as a state-of-the-art method which is comparably robust against motion artifacts. Participants performed basketball dribbling in short blocks of 10 sec either with their dominant right hand (DH) or with their non-dominant left hand (NDH) according to their group. Simultaneously, hemodynamic changes were assessed in bilateral sensorimotor cortex, whereas creative performance was tested before and after the basketball dribbling task. While the results revealed typical hemodynamic activation pattern during basketball dribbling, no effects on creative performance were found. Consequently, since not all hypotheses could be confirmed, the limitations of this study were discussed appropriately and in detail.

All in all, this study is of interest as it systematically extends previous findings on cortical activity during complex movements. The manuscript is well written, the study design is clear and comprehensible, and the authors applied state-of-the-art data assessment and analysis methods.

Hence, I only have some minor comments, which might be considered before publication. See below for some specific suggestions for potential improvements, which you may find useful (the order corresponds to the manuscript and not to prioritization):

1. L. 119-145: The aim of the present study is made clear and comprehensible in this paragraph. However, to improve readability, you might consider splitting this into separate paragraphs of aim and introduction to fNIRS. Moreover, the direct advantages of FNIRS over other techniques (EEG, fMRI) could be more pronounced here.

2. L. 135: The description of the typical fNIRS signal is absolutely correct, but you might add that the decrease in deoxy-Hb occurs at a lower rate than the increase of oxy-Hb.

3. L. 151-157: The aim at predicting the intervention group membership based on SMC activation pattern could be motivated and explained more profoundly as it is not clear enough why this analysis should reveal promising results.

4. L. 158: The abbreviation “DT” is mentioned here but has not been introduced before.

5. L. 165-174: The description of your sample should include the fact that you also assessed the participant’s experience in (basket)ball dribbling as well as general physical activity habits. Please consider adding this here.

6. L. 177-178: Did you control for dribbling speed during the 10 sec dribbling periods, for example using a metronome? If not, do you think that the number of dribbles (e.g., one participant dribbling the ball 5x vs. another participant dribbling the ball 15x) might impact your results?

7. L. 179-180: Please explain why you implemented randomized resting periods of different lengths.

8. L. 260: Please change “trails” into “trials”.

9. L. 303: Please use the abbreviation “fNIRS” consistently throughout the manuscript (see also l. 590, l. 612 and figure 2).

10. L. 330-331: The term “Sensorimotor function” is mentioned here first and the distinction between motor vs. sensation has not been introduced before. Please consider introducing this in the paragraph describing your regions of interest.

11. L. 353: Please indicate the nature of the concentration changes, i.e., increase or decrease (this also refers to further passages in this paragraph).

12. L. 543-552: Do you think that randomizing the order of verbal and figural DT task (instead of always having the same order) could have led to other results as well?

13. Figure 3: Please consider indicating significant differences here.

Reviewer #2: The manuscript is very well written and has high quality both in terms of content and grammar, which makes the text fluent and helps the audience understand a lot. In my view there are a slightly concerns which are presented below.

Introduction

Line 158, P 7: What word do the abbreviations DT represent?

Method

Line 216, P 16: The digit of 2 must write with letter.

Line 280, P 12: It seems in phrase of “has shown to by a reliable method …” the word of “by” is applied wrong. Instead of it must use “be”.

Reviewer #3: REVIEW Sensorimotor cortex activity during basketball dribbling and its relation to creativity

Major Revision:

The aim of the study was to examine whether creative performance is enhanced by a prior increase in the level of activity in one of the cerebral hemispheres (induced by unilateral dribbling movements). Furthermore, the authors studied hemodynamic response alterations during basketball dribbling with the dominant right and the non-dominant left hand.

Results show that right hand dribbling led to stronger contralateral compared to ipsilateral activation, whereas left hand dribbling led to a more bilateral activation pattern. Thus, the results confirm previous studies that have shown comparable findings in various motor tasks. However, the authors were unable to replicate the positive effects of unilateral hand movements on creative performance.

In summary, this is a very interesting study that provides profound insights into the effects of targeted motor cortex preactivation on creative performance and hemodynamic response alterations during basketball dribbling. Nevertheless, I have some methodological comments. I strongly recommend reanalysis of the fNIRS-data.

P8 L178 “Participants had to dribble a standard (size 7) basketball 16 times for 10 sec each, while sitting on a chair.”

The frequency of the movement has a significant influence on the hemodynamic response alterations. How was the dribbling frequency realized and secured? With a metronome? Furthermore, the execution of the movement also has a decisive influence on brain processing. Was the execution observed? (I assume that there were differences between the execution with the dominant and the non-dominant hand. Especially in novices, e.g., high/low dribbling?) The authors discuss that the different perception of task difficulty has an influence. The influence of the different movement executions itself should be added when discussing the different results.

P11 L256 “The enPruneChannels function was applied to exclude channels with low signal-to-noise ratio (dRange: -1e+04 / 1e+04, SNRthresh: 2, SDrange: 0.0 / 45.0, reset: 0) but no such channels were identified.”

I am not surprised that no channel with low signal-to-noise (SNR) ratio was identified. The selected threshold SNRthresh: 2 is too low. As a result, channels with a low SNR also pass this data quality check.

Explanation: The input SNRthresh is decisive. The Homer2 default setting SNRthresh: 2 has already been adopted by a number of fNIRS studies. However, SNRthresh: 2 corresponds to a coefficient of variation (CV) of 50%! fNIRS analysis default values for CV are 10% (strict) or 15% (less strict). 10% CV corresponds to SNRthresh: 10 and 15% CV corresponds to SNRthresh: 6.67. In the Homer forum Meryem Yücel recommends SNRthresh 5. This value is even less strict. I recommend to reanalyze the data using 6.67 for SNRthresh. In general, CV and SNR behave reciprocally (CV = std(d)/mean(d). SNR = mean(d)/std(d)). The higher I choose SNRthresh, the stricter I set the threshold. The equation for converting CV and SNR is: SNRthresh = 1/CV * 100 (or 100/CV).

P12 L267 “General hemodynamic drift, as measured by the short-distance channels, was regressed out of the hemodynamic response function (HRF) by using hmrDeconvHRF_DriftSS ([…], flagSSmethod: 0, […])”

FlagSSmethod: 0 means short separation regression is performed with the nearest short separation channel. David Boas recommends using flagSSmethod - 1 instead. Here, short separation regression is performed with the short separation channel with the greatest correlation. Theoretically, flagSSmethod - 0 makes the most sense. However, Boas points out that in practice noisy short separation channels can cause big problems and therefore flagSSmethod - 1 should be used.

P13 Table1 In each ROI, three channels were included. However, there are four pre- and postcentral channels in the left hemisphere. According to which criterion was Ch 6 not included in the ROI? The same question arises for channel 10 (Postcentral_L). Because of the symmetry? To better illustrate the configuration, the anatomical positions should be added in Figure 1C.

P14 L299 Participants were asked “Do you have experience with sports that involve dribbling a ball (e.g., basketball, handball, etc.)?” (Yes/No) The groups were in terms of specific motor expertise. NDH (12/9), DH (11/11)

The authors did balance the groups in terms of specific motor expertise. Nevertheless, if possible, the specific motor expertise should be described in more detail. I recommend, for example, the classification scheme according to McKay et al. 2022 ("Defining Training and Performance Caliber: A Participant Classification Framework"). If subjects differ on more accurate classification, the differences in expertise could have a huge impact on the task-related brain processing observed here.

6. PLOS authors have the option to publish the peer review history of their article (what does this mean?). If published, this will include your full peer review and any attached files.

Reviewer #1: No

Reviewer #2: No

Reviewer #3: **Yes: **Dr. Daniel Carius

---

## [Author Response · Author response to Decision Letter 0]

1 Mar 2023

Response to Reviewers

Concerning the revision of the manuscript Sensorimotor cortex activity during basketball dribbling and its relation to creativity (PONE-D-22-32438)

Dear reviewers,

Dear editors of PLOS ONE, 

In the name of all my co-authors, I would like to thank you for your thoughtful and constructive feedback for our manuscript. We hereby present to you our responses to all points raised during your reviews and the subsequent changes we made to the manuscript. 

Concerning the document ‘Revised Manuscript with Track Changes’ I must clarify that we started revision of the manuscript using the ‘Track Changes’-function in Microsoft Word, where every change done in the document is being tracked. However, after near completion of the revision using this method, we encountered a (as we learned very common) bug in Word, where the software is unable to keep functioning properly, due to there being very many changes in the document. As this software bug in Word rendered further revision as well as navigation through the document immensely slow, we decided to switch to another method for revising the manuscript. Therefore, the document ‘Revised Manuscript with Track Changes’ is a version of the new and revised manuscript where we highlighted all major changes to the manuscript. It is possible that some very minor corrections, such as spelling errors or comma placements, were performed as well but may not be highlighted in that document. For that we want to apologize but we hope that you find our fallback solution for this revision equally as informative. 

Kind regards, 

Thomas Kanatschnig

Reviewer 1

General review: 

03-01-2023 

Review of PONE-D-22-32438

Sensorimotor cortex activity during basketball dribbling and its relation to creativity

Thank you for the opportunity to review this manuscript. The current article is an original research article on sensorimotor cortex activity during basketball dribbling and its relation to creativity. For that purpose, the authors applied functional near-infrared spectroscopy (fNIRS) as an established method to measure hemodynamic changes in selected brain regions. The aim of this study was to replicate previous findings on enhanced creative performance by increasing the level of activity in one of the brains hemispheres through unilateral hand movements. The novel aspects of this study can be found in the incorporation of basketball dribbling as a more advanced motor task and the application of fNIRS as a state-of-the-art method which is comparably robust against motion artifacts. Participants performed basketball dribbling in short blocks of 10 sec either with their dominant right hand (DH) or with their non-dominant left hand (NDH) according to their group. Simultaneously, hemodynamic changes were assessed in bilateral sensorimotor cortex, whereas creative performance was tested before and after the basketball dribbling task. While the results revealed typical hemodynamic activation pattern during basketball dribbling, no effects on creative performance were found. Consequently, since not all hypotheses could be confirmed, the limitations of this study were discussed appropriately and in detail.

All in all, this study is of interest as it systematically extends previous findings on cortical activity during complex movements. The manuscript is well written, the study design is clear and comprehensible, and the authors applied state-of-the-art data assessment and analysis methods.

Hence, I only have some minor comments, which might be considered before publication. See below for some specific suggestions for potential improvements, which you may find useful (the order corresponds to the manuscript and not to prioritization):

Response:

Dear reviewer 1,

Thank you for your favorable comments to our manuscript and research. You brought up important aspects, which helped us improve our manuscript substantially. Here are the points we adapted following your feedback.

Comment 1: 1. L. 119-145: The aim of the present study is made clear and comprehensible in this paragraph. However, to improve readability, you might consider splitting this into separate paragraphs of aim and introduction to fNIRS. Moreover, the direct advantages of FNIRS over other techniques (EEG, fMRI) could be more pronounced here.

L. 130-147: We divided the mentioned paragraph into two separate paragraphs, the second of which is now specifically concentrated on the fNIRS technology. Furthermore, we added information about the advantages that fNIRS brings compared to other neuroimaging techniques (new reference [31]).

Comment 2: 2. L. 135: The description of the typical fNIRS signal is absolutely correct, but you might add that the decrease in deoxy-Hb occurs at a lower rate than the increase of oxy-Hb.

L. 135-136: We added this aspect.

Comment 3: 3. L. 151-157: The aim at predicting the intervention group membership based on SMC activation pattern could be motivated and explained more profoundly as it is not clear enough why this analysis should reveal promising results.

L. 163-167: A more detailed description for the motivation to perform linear discriminant analysis on the data was inserted, including additional references from research endeavors in the field of rehabilitation for stroke (new references [37,38]) and dysphagia (new reference [39]) using fNIRS.

Comment 4: 4. L. 158: The abbreviation “DT” is mentioned here but has not been introduced before.

L. 168: The missing meaning of the abbreviation DT (divergent thinking) was inserted.

Comment 5: 5. L. 165-174: The description of your sample should include the fact that you also assessed the participant’s experience in (basket)ball dribbling as well as general physical activity habits. Please consider adding this here.

L. 178-180: A sentence concerning the assessment of general sports-related habits was added.

Comment 6: 6. L. 177-178: Did you control for dribbling speed during the 10 sec dribbling periods, for example using a metronome? If not, do you think that the number of dribbles (e.g., one participant dribbling the ball 5x vs. another participant dribbling the ball 15x) might impact your results?

L. 194-201 and 642-647: In response to this question, yes, we do think that generally the frequency of dribbling has an impact on the hemodynamic response. However, considering that all our participants were given only little leeway in the pacing of the dribbling, we do not think that they differed greatly in that aspect. We must say that we did not use a method to control or measure the dribbling frequency of the participants, such as a metronome or counting the number of dribbles. Using a metronome to specify the dribbling frequency would have been possible. However, we assumed that that could have made the task more difficult for participants who are not as experienced with ball dribbling, which was not our intention. All participants were instructed equally. This paragraph explains the instructions and was added to the manuscript (L. 194-201): 

“We requested each participant to dribble the ball either with their dominant right (DH) or non-dominant left (NDH) hand, depending on group assignment. Each participant should dribble the basketball in a way that they felt comfortable during each of the 16 ten-second-trials, while sitting upright on a chair with no wheels or armrests, ensuring the necessary stability and freedom of movement. They were not allowed to stop dribbling midway through a trial and were asked always to dribble the ball up to approximately the height of their waist. They were also allowed to look at the ball while dribbling to have better control over it”. 

We wanted each participant to be able to have some control over the pace of the dribbling themselves but considering that they were not allowed to stop dribbling during a trial, the average number of dribbles per trial must have been quite similar across participants. Nevertheless, the lack of a measurement of dribbling frequency can be considered to be a shortcoming of our experimental design, which is why we also added a paragraph about this issue in the ‘Limitations’ section (L. 642-647).

Comment 7: 7. L. 179-180: Please explain why you implemented randomized resting periods of different lengths.

L. 192-194: We decided to implement randomized resting periods (also referred to as ‘jittering’) to accommodate for the influence of slow hemodynamic fluctuations, i.e., Mayer waves, in the cortical blood flow. This is an easy but effective method that is often used in fNIRS and fMRI experiments to tackle the problem of physiological noise in the hemodynamic response. Additional references are provided that cover the topic of Mayer waves in fNIRS analysis (new reference [41]) and an example of an fMRI study that used a similar approach (new reference [42]).

Comment 8: 8. L. 260: Please change “trails” into “trials”.

L. 293: This was corrected. 

Comment 9: 9. L. 303: Please use the abbreviation 'fNIRS” consistently throughout the manuscript (see also l. 590, l. 612 and figure 2).

L. 342, 355, 664 and Fig 2: The consistent use of the term ‘fNIRS’ was established throughout the whole manuscript.

Comment 10: 10. L. 330-331: The term “Sensorimotor function” is mentioned here first and the distinction between motor vs. sensation has not been introduced before. Please consider introducing this in the paragraph describing your regions of interest.

L. 324-328: A description of the term ‘Sensorimotor Function’ in the context of our regions of interest was added.

Comment 11: 11. L. 353: Please indicate the nature of the concentration changes, i.e., increase or decrease (this also refers to further passages in this paragraph).

L. 397-463: Increases and decreases in hemoglobin concentrations were formulated more clearly throughout the whole section ‘Sensorimotor cortex activation’.

Comment 12: 12. L. 543-552: Do you think that randomizing the order of verbal and figural DT task (instead of always having the same order) could have led to other results as well?

L. 623-627: Yes, the order of task presentation (verbal/figural) could have had an impact on the participant's task performance. We discuss this point in the ‘Limitations’ section and added additional considerations.

Comment 13: 13. Figure 3: Please consider indicating significant differences here.

L. 434 and Fig 3: Statistically significant differences are now indicated in Fig 3.

Reviewer 2

General review:

The manuscript is very well written and has high quality both in terms of content and grammar, which makes the text fluent and helps the audience understand a lot. In my view there are a slightly concerns which are presented below.

Response:

Dear reviewer 2,

Thank you for your feedback and considerations, which we were happy to implement in our manuscript.

Comment 1: Line 158, P 7: What word do the abbreviations DT represent?

L. 168: As also mentioned by reviewer 1, the missing meaning of the abbreviation DT (divergent thinking) was inserted.

Comment 2: Line 216, P 16: The digit of 2 must write with letter.

L. 237: This was corrected.

Comment 3: Line 280, P 12: It seems in phrase of “has shown to by a reliable method …” the word of “by” is applied wrong. Instead of it must use “be”.

L. 312: This was corrected.

Reviewer 3

General review:

REVIEW Sensorimotor cortex activity during basketball dribbling and its relation to creativity

Major Revision:

The aim of the study was to examine whether creative performance is enhanced by a prior increase in the level of activity in one of the cerebral hemispheres (induced by unilateral dribbling movements). Furthermore, the authors studied hemodynamic response alterations during basketball dribbling with the dominant right and the non-dominant left hand.

Results show that right hand dribbling led to stronger contralateral compared to ipsilateral activation, whereas left hand dribbling led to a more bilateral activation pattern. Thus, the results confirm previous studies that have shown comparable findings in various motor tasks. However, the authors were unable to replicate the positive effects of unilateral hand movements on creative performance.

In summary, this is a very interesting study that provides profound insights into the effects of targeted motor cortex preactivation on creative performance and hemodynamic response alterations during basketball dribbling. Nevertheless, I have some methodological comments. I strongly recommend reanalysis of the fNIRS-data.

Response:

Dear Dr. Daniel Carius,

Thank you very much for your thorough analysis of our manuscript and methodology. The points you raised in your review led us to reanalyze the fNIRS data. We go into further detail regarding these analysis aspects in the respective detailed responses to comments 2 and 3 below. Please also see our responses to the remaining comments of your review below.

Comment 1: P8 L178 “Participants had to dribble a standard (size 7) basketball 16 times for 10 sec each, while sitting on a chair.” The frequency of the movement has a significant influence on the hemodynamic response alterations. How was the dribbling frequency realized and secured? With a metronome? Furthermore, the execution of the movement also has a decisive influence on brain processing. Was the execution observed? (I assume that there were differences between the execution with the dominant and the non-dominant hand. Especially in novices, e.g., high/low dribbling?) The authors discuss that the different perception of task difficulty has an influence. The influence of the different movement executions itself should be added when discussing the different results.

L. 194-201 and 642-647: As discussed above under comment 6 of reviewer 1, yes, we do think that generally the frequency of dribbling has an impact on the hemodynamic response. However, considering that all our participants were given only little leeway in the pacing of the dribbling, we do not think that they differed greatly in that aspect. We must say that we did not use a method to control or measure the dribbling frequency of the participants, such as a metronome or counting the number of dribbles. Using a metronome to specify the dribbling frequency would have been possible. However, we assumed that that could have made the task more difficult for participants who are not as experienced with ball dribbling, which was not our intention. All participants were instructed equally. This paragraph explains the instructions and was added to the manuscript (L. 194-201): 

“We requested each participant to dribble the ball either with their dominant right (DH) or non-dominant left (NDH) hand, depending on group assignment. Each participant should dribble the basketball in a way that they felt comfortable during each of the 16 ten-second-trials, while sitting upright on a chair with no wheels or armrests, ensuring the necessary stability and freedom of movement. They were not allowed to stop dribbling midway through a trial and were asked always to dribble the ball up to approximately the height of their waist. They were also allowed to look at the ball while dribbling to have better control over it”. 

We wanted each participant to be able to have some control over the pace of the dribbling themselves but considering that they were not allowed to stop dribbling during a trial, the average number of dribbles per trial must have been quite similar across participants. Nevertheless, the lack of a measurement of dribbling frequency can be considered to be a shortcoming of our experimental design, which is why we also added a paragraph about this issue in the ‘Limitations’ section (L. 642-647).

Comment 2: P11 L256 “The enPruneChannels function was applied to exclude channels with low signal-to-noise ratio (dRange: -1e+04 / 1e+04, SNRthresh: 2, SDrange: 0.0 / 45.0, reset: 0) but no such channels were identified.” I am not surprised that no channel with low signal-to-noise (SNR) ratio was identified. The selected threshold SNRthresh: 2 is too low. As a result, channels with a low SNR also pass this data quality check. Explanation: The input SNRthresh is decisive. The Homer2 default setting SNRthresh: 2 has already been adopted by a number of fNIRS studies. However, SNRthresh: 2 corresponds to a coefficient of variation (CV) of 50%! fNIRS analysis default values for CV are 10% (strict) or 15% (less strict). 10% CV corresponds to SNRthresh: 10 and 15% CV corresponds to SNRthresh: 6.67. In the Homer forum Meryem Yücel recommends SNRthresh 5. This value is even less strict. I recommend to reanalyze the data using 6.67 for SNRthresh. In general, CV and SNR behave reciprocally (CV = std(d)/mean(d). SNR = mean(d)/std(d)). The higher I choose SNRthresh, the stricter I set the threshold. The equation for converting CV and SNR is: SNRthresh = 1/CV * 100 (or 100/CV).

L. 286-289, 380, 397-463 and 554: We followed the advice to rerun the analysis of the fNIRS data using a stricter value for the SNR threshold in the ‘enPruneChannels’ function of Homer2. We concluded that adjusting the SNR threshold was a crucial addition to our pre-processing pipeline to verify that our signal quality was sufficient. The reanalysis revealed that only four channels in the whole dataset (meaning four channels in total, out of the sum of all short-separation channels from all participants taken together) had a signal-to-noise ratio lower than 6.67, which was the new recommended threshold (L. 286-289). All those channels were short-separation channels, no long-separation channel showed a SNR lower than 6.67. All four short-separation channels with a SNR lower than 6.67 were excluded from further analyses by Homer2. Considering that only four channels did not meet this criterion, we presume that the signal quality in our dataset was adequate. Consequently, all statistical analyses including the fNIRS data were rerun again as well. For the reanalysis we switched from SPSS version 27 to the newest version 29 (L. 380). All results presented in the manuscript were adapted accordingly. The results changed slightly but not significantly. All previous significant and non-significant effects remained the same (L. 397-463 and 554).

Comment 3: P12 L267 “General hemodynamic drift, as measured by the short-distance channels, was regressed out of the hemodynamic response function (HRF) by using hmrDeconvHRF_DriftSS ([…], flagSSmethod: 0, […])” FlagSSmethod: 0 means short separation regression is performed with the nearest short separation channel. David Boas recommends using flagSSmethod - 1 instead. Here, short separation regression is performed with the short separation channel with the greatest correlation. Theoretically, flagSSmethod - 0 makes the most sense. However, Boas points out that in practice noisy short separation channels can cause big problems and therefore flagSSmethod - 1 should be used.

Thank you for the advice concerning the ‘flagSSmethod’ parameter. We decided it would be more reasonable to keep the parameter ‘flagSSmethod’ in the ‘hmrDeconvHRF_DriftSS’ function set to ’0’ (nearest short-separation channel), as originally intended, and not setting it to ‘1’ (short-separation channel with the greatest correlation), as it would be the advice from Boas et al. for noisy data. There are two main reasons for our decision. First, as discussed under comment 2, the reanalysis of our data revealed that only four short-separation channels in the dataset showed a SNR value lower than 6.67. These channels were excluded from the analysis. All other short- as well as long-separation channels showed adequate signal quality. Therefore, we presume that noise was not a severe issue in our data and consequently that the suggestion of Boas et al. does not necessarily apply in our case. Second, as you also mentioned, the value ‘0’ (nearest short-separation channel) would theoretically make the most sense in this scenario of data analysis. This way Homer2 will always choose a short-separation channel that is nearest to the area where the physiological noise in a specific long-separation channel stems from, rather than choosing the one which has the highest correlation but might be spatially quite distant from the long-separation channel in question. This is a deliberation that we followed from the beginning of data analysis. Therefore, it would be contradictory to our original intentions to change this parameter. We hope we could hereby sufficiently explain our position regarding this aspect. 

Comment 4: P13 Table1 In each ROI, three channels were included. However, there are four pre- and postcentral channels in the left hemisphere. According to which criterion was Ch 6 not included in the ROI? The same question arises for channel 10 (Postcentral_L). Because of the symmetry? To better illustrate the configuration, the anatomical positions should be added in Figure 1C.

L. 273-282 and Fig 1: As you describe correctly, the reason why we chose not to include channels 6 and 10 into the ROIs ‘Precentral Left’ and ‘Postcentral Left’, respectively, was to establish symmetry. We decided it would be favorable to only analyze the same corresponding areas when comparing the brain activation between the left and right hemisphere. Following your advice, the anatomical positions of the fNIRS optodes were added in Fig 1 for clarity. The abbreviations for sources, detectors and channels were changed to lowercase letters to prevent confusion with the newly added EEG labels. Colored highlighting was added to show the ROIs.

Comment 5: P14 L299 Participants were asked “Do you have experience with sports that involve dribbling a ball (e.g., basketball, handball, etc.)?” (Yes/No) The groups were in terms of specific motor expertise. NDH (12/9), DH (11/11). The authors did balance the groups in terms of specific motor expertise. Nevertheless, if possible, the specific motor expertise should be described in more detail. I recommend, for example, the classification scheme according to McKay et al. 2022 ("Defining Training and Performance Caliber: A Participant Classification Framework"). If subjects differ on more accurate classification, the differences in expertise could have a huge impact on the task-related brain processing observed here.

We agree that the individual level of expertise in ball dribbling of our subjects probably affected their sensorimotor activation during the dribbling task. A more profound method to classify the level of dribbling expertise and general athleticism, such as the guidelines of McKay et al. (2022), would have been a beneficial addition to control for such effects. Unfortunately, we did not gather detailed information to which extend our subjects spent time on physical activities on a weekly basis, which is why we are not be able to retrospectively classify them according to these guidelines. However, considering that we recruited subjects from the general population (not athletes in particular) and that according to McKay et al. (2022) the majority of the global population are either sedentary (tier 0) or only recreationally active (tier 1) we presume that our data was not strongly confounded by outliers with very high aptitude in ball dribbling. 

Additional changes

• All Headings were reformatted following the PLOS ONE formatting guidelines.

• L. 296: Reference to Figs 4 and 5 was removed, for the sake of continuity in figure numbering.

• All figures were reworked and reformatted following the PLOS ONE formatting guidelines.

• All figure legends were reformatted and moved to their respective locations following the PLOS ONE formatting guidelines.

• All Tables were reformatted following the PLOS ONE formatting guidelines.

• Supporting information captions were relocated to the end of the manuscript and reformatted. Mentions of supporting files were reformatted as well.

• The funding statement "The authors acknowledge the financial support by the University of Graz." was removed from this section, as pointed out by the editor.

• The sections ‘Author contributions’, ‘Data availability’, ‘Competing interests’ and ‘Figure Legends’ were removed from the manuscript.

---

## [Decision Letter · Decision Letter 1]

27 Mar 2023

Sensorimotor cortex activity during basketball dribbling and its relation to creativity

PONE-D-22-32438R1

Dear Dr. Kanatschnig,

We’re pleased to inform you that your manuscript has been judged scientifically suitable for publication and will be formally accepted for publication once it meets all outstanding technical requirements.

Kind regards,

Noman Naseer, PhD

Academic Editor

PLOS ONE

Additional Editor Comments (optional):

Reviewers are satisfied with the revisions made. 

Reviewers' comments:

Reviewer's Responses to Questions

**Comments to the Author**

1. If the authors have adequately addressed your comments raised in a previous round of review and you feel that this manuscript is now acceptable for publication, you may indicate that here to bypass the “Comments to the Author” section, enter your conflict of interest statement in the “Confidential to Editor” section, and submit your "Accept" recommendation.

Reviewer #2: All comments have been addressed

Reviewer #3: All comments have been addressed

2. Is the manuscript technically sound, and do the data support the conclusions?

Reviewer #2: Yes

Reviewer #3: Yes

3. Has the statistical analysis been performed appropriately and rigorously? 

Reviewer #2: Yes

Reviewer #3: Yes

4. Have the authors made all data underlying the findings in their manuscript fully available?

Reviewer #2: Yes

Reviewer #3: Yes

5. Is the manuscript presented in an intelligible fashion and written in standard English?

Reviewer #2: Yes

Reviewer #3: Yes

6. Review Comments to the Author

Reviewer #2: (No Response)

Reviewer #3: I think that the authors have sufficiently implemented my recommendations and congratulate them in advance on their publication. I can very well understand the authors' responses to my comments. Also, the crucial methodological recommendations were incorporated and reanalyses were performed.

7. PLOS authors have the option to publish the peer review history of their article (what does this mean?). If published, this will include your full peer review and any attached files.

Reviewer #2: No

Reviewer #3: **Yes: **Dr. Daniel Carius

---

## [Editor Report · Acceptance letter]

30 Mar 2023

PONE-D-22-32438R1 

Sensorimotor cortex activity during basketball dribbling and its relation to creativity 

Dear Dr. Kanatschnig:

I'm pleased to inform you that your manuscript has been deemed suitable for publication in PLOS ONE. Congratulations! Your manuscript is now with our production department. 

Kind regards, 

on behalf of

Dr. Noman Naseer 

Academic Editor

PLOS ONE